# GCR1 Positively Regulates UV-B- and Ethylene-Induced Stomatal Closure via Activating GPA1-Dependent ROS and NO Production

**DOI:** 10.3390/ijms23105512

**Published:** 2022-05-15

**Authors:** Xue Li, Qi Fu, Fu-Xing Zhao, Yi-Qing Wu, Teng-Yue Zhang, Zhong-Qi Li, Jun-Min He

**Affiliations:** School of Life Sciences, Shaanxi Normal University, Xi’an 710119, China; lx1165930583@163.com (X.L.); defind@126.com (Q.F.); fuxing1031@sina.com (F.-X.Z.); bamboo9606@163.com (Y.-Q.W.); zhangtengyue@snnu.edu.cn (T.-Y.Z.)

**Keywords:** *Arabidopsis thaliana*, ethylene, G protein-coupled receptor GCR1, Gα subunit GPA1, nitric oxide, reactive oxygen species, stomatal closure, UV-B

## Abstract

Heterotrimeric G proteins function as key players in guard cell signaling to many stimuli, including ultraviolet B (UV-B) and ethylene, but whether guard cell G protein signaling is activated by the only one potential G protein-coupled receptor, GCR1, is still unclear. Here, we found that *gcr1* null mutants showed defects in UV-B- and ethylene-induced stomatal closure and production of reactive oxygen species (ROS) and nitric oxide (NO) in guard cells, but these defects could be rescued by the application of a Gα activator or overexpression of a constitutively active form of Gα subunit GPA1 (cGPA1). Moreover, the exogenous application of hydrogen peroxide (H_2_O_2_) or NO triggered stomatal closure in *gcr1* mutants and *cGPA1* transgenic plants in the absence or presence of UV-B or ethylene, but exogenous ethylene could not rescue the defect of *gcr1* mutants in UV-B-induced stomatal closure, and *gcr1* mutants did not affect UV-B-induced ethylene production in Arabidopsis leaves. These results indicate that GCR1 positively controls UV-B- and ethylene-induced stomatal closure by activating GPA1-dependent ROS and NO production in guard cells and that ethylene acts upstream of GCR1 to transduce UV-B guard cell signaling, which establishes the existence of a classic paradigm of G protein signaling in guard cell signaling to UV-B and ethylene.

## 1. Introduction

Ligand signaling through G protein-coupled receptors (GPCRs) is a widespread mechanism for extracellular signal perception in eukaryotic organisms. In the classic paradigm of G protein signaling, the GPCR associates with heterotrimeric G proteins composed of α, β, and γ subunits. Ligand binding to the GPCR causes Gα to exchange GDP for GTP, leading to the dissociation of the G protein complex into Gα-GTP and Gβγ dimers. These two functional subunits interact with a variety of downstream effectors to activate different signaling cascades. The intrinsic GTPase activity of Gα, which can be accelerated by the regulator of G protein signaling (RGS) proteins, returns the G protein to the inactive trimeric state [1].

Plant G protein signal transduction is modeled on the well-established animal paradigm, but many experimental pieces of evidence show that plant G protein signaling has taken a very different evolutionary path. Most importantly, while in animal systems G protein signaling is activated by thousands of identified GPCRs, only one canonical GPCR gene, named *GCR1*, was identified by homology searches of the EST databases in *Arabidopsis*
*thaliana* [2,3]. Additional GPCR candidates were mined based on their transmembrane domains [4,5,6], with a list of up to 56 numbers, but of which GCR1 was bioinformatically considered to be the best candidate based on GPCR fold analysis [7]. Despite being the most studied potential candidate, the role of GCR1 as a GPCR remains biochemically unproven, due to the lack of an identified ligand that binds GCR1 and the lack of demonstration of guanine nucleotide exchange factors (GEFs) activity [5,8]. Therefore, it remains an unsettled controversy whether these GPCRs in animal models are present in plants [7,9]. Similarly, compared to the animal kingdom, where multiple *G**α*, *G**β*, *G**γ*, and *RGS* genes exist, there is only one canonical G protein *α*-subunit1 (*GPA1*) [10], three extra-large GTP-binding proteins (XLG1, XLG2, and XLG3) [11], one Arabidopsis G protein *β*-subunit1 (*AGB1*) [12], three Arabidopsis G protein *γ*-subunit (*AGG1*, *AGG2*, and *AGG3*) [13,14], and one *RGS* (*RGS1*) [15] genes in Arabidopsis. However, consistent with that GCR1 functions as a GPCR, studies have shown that GCR1 physically interacts with GPA1 in a manner dependent on the intracellular domains of GCR1 [8], as has been shown for mammalian GPCRs [16]. Furthermore, *gcr1* mutants exhibit a subset of the phenotypes seen in *gpa1* mutants, such as on the cell cycle and the ABA sensitivity of seed germination [8,17,18,19,20]. In contrast, compared to animal Gαs, evidence has shown that GPA1 has a very low GTPase activity and spontaneously exchanges GDP for GTP without any involvement of GPCRs, suggesting that Gα subunits in plants are constitutively active [21]. Moreover, plant G proteins have been proven to physically interact with atypical receptors, such as the Arabidopsis RGS1 and some receptor-like kinases (RLKs) [15,22,23,24,25,26,27,28,29,30,31]. Thus, it has been proposed that, instead of the GPCR-dependent activation of G proteins employed in animals, plant G proteins are activated by phosphorylation or by signals inhibiting the deactivation of a constitutively active Gα [32,33]. Nevertheless, GCR1 satisfies the essential criteria for the identification of a protein as a GPCR. However, whether plant G protein signaling is also activated by GCR1 is still worthwhile to be researched.

Guard cells respond to many stimuli to regulate the stomatal aperture to an optimal level and thus optimize plant growth by controlling water loss, gas exchange, and innate immunity. Arabidopsis T-DNA insertional null mutants of the sole canonical Gα subunit, *GPA1*, are insensitive to stomata closing induced by sphingosine-1-P (S1P) [34,35], phosphatidic acid (PA) [36], extracellular calmodulin (ExtCaM) [37,38], extracellular ATP [39], ultraviolet B (UV-B) radiation [40], ethylene [41], brassinosteroid (BR) [42], and the pathogen-associated molecular pattern flg22 [43], as well as to abscisic acid (ABA) and S1P inhibition of stomatal opening [34,35,44,45], suggesting that plant Gα subunit GPA1 responds to various stimuli as a key regulator of stomatal signaling. Further studies indicate that GPA1 mediates several stimuli-regulated stomatal movements by inducing the production of reactive oxygen species (ROS) and nitric oxide (NO) and regulating the activities of inward K^+^ channels and slow anion channels in guard cells [37,38,39,40,41,42,44,45]. Despite the progress achieved in our understanding of the role of G proteins in stomatal signaling, it is still unclear how multiple stimuli activate G protein signaling and whether GCR1 is involved in the activation of G protein signaling in guard cells. To address this problem, Pandey and Assmann [8] compared the ABA and S1P responses between *gcr1* and *gpa1* mutant guard cells and found that the *gcr1* mutants exhibited hypersensitivity in ABA and S1P inhibition of stomatal opening and promoting of stomatal closure, whereas the *gpa1* mutants exhibited insensitivity in ABA and S1P inhibition of stomatal opening and normal response in ABA and S1P promoting of stomatal closure [8,34,44]. To reconcile these observations, the authors hypothesized that GCR1 negatively regulates ABA signaling via a mechanism dependent on its direct binding with and negatively regulating GPA1 or independent of its binding to GPA1 [8]. However, whether GCR1 is required for guard cell signaling to other stimuli is uncertain. If this is the case, the next question is whether and how it regulates G protein signaling in guard cells. All these problems are still unclear and worth studying.

In this study, we used Arabidopsis *gcr1* null mutants and the transgenic lines overexpressing a constitutively active form of *GPA1* (*c**GPA1*) in wild-type and *gcr1* mutants to show that GCR1 positively mediates ethylene- and UV-B-induced ROS and NO production in guard cells and subsequent stomatal closure in a GPA1-dependent manner. Our results establish that a classic paradigm of G protein signaling exists in the guard cell responses to ethylene and UV-B.

## 2. Results

### 2.1. GCR1 Positively Regulates UV-B- and Ethylene-Induced Stomatal Closure

In this study, based on the previous finding that heterotrimeric G protein α subunit GPA1 mediates stomatal closure in *Arabidopsis* leaves induced by 0.5 W/m^2^ UV-B radiation or 100 μmol/L 1-aminocyclopropane-1-carboxylic acid (ACC: the direct precursor of ethylene during its biosynthesis) [40,41], we used 0.5 W/m^2^ UV-B and 100 μmol/L ACC as UV-B and ethylene treatments, respectively, and further explored the role of the sole prototypical Arabidopsis GPCR gene, *GCR1*, in UV-B- and ethylene-induced stomatal closure in Arabidopsis. When freshly prepared Arabidopsis leaves with open stomata were exposed to light alone or with 0.5 W/m^2^ UV-B radiation for 3 h, UV-B induced stomatal closure in the wild-type (WT), but could not trigger the closing of stomata in the *gcr1-1* and *gcr1-2* mutants (Figure 1A). Similarly, when the leaves were treated with 100 μmol/L ACC for 3 h, ACC induced stomatal closure in the WT, but this effect of ACC was defective in the *gcr1-1* and *gcr1-2* mutants (Figure 1B). These genetic results indicate that GCR1 acting as a positive regulator mediates both UV-B- and ethylene-induced closing of stomata.

The size of stomatal pores controls water loss. Thus, we posited that UV-B or ACC treatment reduces water loss and that this effect is dependent on GCR1. When leaves of WT and *gcr1* mutants treated with or without 0.5 W/m^2^ UV-B or 100 μmol/L ACC were placed under light without MES buffer for 2 h, the UV-B- and ACC-treated WT leaves lost less water than the untreated WT leaves, while UV-B and ACC treatments did not decrease the water loss from the *gcr1* mutant leaves (Appendix A). These findings further confirmed the effect of UV-B and ethylene on stomatal aperture in WT and *gcr1* leaves.

Our finding that GCR1 acts as a positive regulator in both UV-B- and ethylene-induced stomatal closure is contrary to the negative role of GCR1 in ABA and S1P inhibition of stomatal opening and promoting of stomatal closure [8]. To further confirm the role of GCR1 in ABA guard cell signaling, we repeated the experiments of Pandey and Assmann [8] in our experimental conditions. Consistent with the results of Pandey and Assmann [8] that *gcr1-3* and *gcr1-4* (two other allelic *GCR1* mutants; Appendix A) exhibited hypersensitivity in ABA inhibition of stomatal opening and promoting of stomatal closure, our results showed that *gcr1-1* and *gcr1-2* were hypersensitive in both ABA responses (Appendix A), which further confirms the results of Pandey and Assmann [8] and supports that GCR1 positively regulates UV-B and ethylene guard cell signaling but negatively controls ABA guard cell signaling.

### 2.2. GCR1 Mediates UV-B- and Ethylene-Induced Stomatal Closure via Activating Gα Subunit GPA1

In the classic paradigm of G protein signaling, heterotrimeric G proteins composed of α, β, and γ subunits link ligand perception by GPCRs with downstream effectors. Because GCR1 and Gα subunit GPA1 interact in plants [8], and both of them play positive roles in guard cell responses to UV-B and ethylene (Figure 1; [40,41]), we further explored whether GCR1 mediates guard cell UV-B and ethylene signaling by activating GPA1. Firstly, we investigated the stomatal responses of the WT Col-0 and mutants *gcr1-1* and *gcr1-2* to Gα activator cholera toxin (CTX) [37]. As shown in Figure 2A,B, CTX not only induced stomatal closure in the leaves of the WT and *gcr1* mutants in the absence of UV-B or ACC but also rescued the defects of the *gcr1-1* and *gcr1-2* mutants in UV-B- and ACC-induced stomatal closure. These results suggest that GCR1 mediates both UV-B- and ethylene-induced stomatal closure via activating Gα. In Arabidopsis, *GPA1* is the only prototypical Gα gene [46]. To further confirm the above pharmacological data, we generated the transgenic plants overexpressing a constitutively active form of GPA1 (cGPA1; GPA1 with a point mutation of Glu-222 to Leu, which locks GPA1 in the active state once activated) [47]. Their overexpression levels of *GPA1* in the leaves of T2 transgenic lines used for further analysis were confirmed by quantitative real-time PCR (qPCR) and revealed no significant difference in the WT vs. *gcr1-1* and *gcr1-2* (Figure 2C). Under our plant growth condition, although the leaves of the cGPA1 transgenic lines were slightly smaller than those of the WT and *gcr1* mutants, no other difference including stomatal density and development was detected between the control plants and the transgenic lines (data not shown). However, the cGPA1 transgenic lines in either the WT Col-0 or *gcr1-1* and *gcr1-2* mutant backgrounds not only exhibited smaller stomatal apertures compared with their corresponding WT and mutants in the absence of UV-B or ACC but also showed significant stomatal closure induced by UV-B and ACC, which is similar to the stomatal response of the WT to UV-B and ACC (Figure 2D). Together, our pharmacological and genetic results convincingly indicate that GCR1 mediates guard cell responses to both UV-B and ethylene by activating Gα subunit GPA1.

### 2.3. GCR1 Mediates UV-B- and Ethylene-Induced Stomatal Closure by Activating GPA1-Dependent ROS Production in Guard Cells

Having established that GCR1 mediates both UV-B- and ethylene-induced stomatal closure by activating Gα subunit GPA1 (Figure 2) and that GPA1 mediates both UV-B- and ethylene-induced stomatal closure via modulating ROS generation in guard cells [40,41], we further explored whether GCR1 mediates both UV-B and ethylene guard cell signaling by activating GPA1-dependent ROS generation. First, we used the ROS-sensitive fluorescent dye 2′,7′-dichlorofluorescin diacetate (H_2_DCFDA) to examine the ROS production in guard cells under the absence or presence of UV-B or ACC. Consistent with the impairment of the *gcr1* mutants in UV-B- and ACC-induced stomatal closure (Figure 1A,B), both the UV-B- and ACC-induced ROS production in guard cells of the WT Col-0 were completely impaired in the guard cells of the *gcr1* mutants (Figure 3), suggesting that GCR1 mediates both UV-B and ethylene stomatal responses by inducing ROS production in guard cells. Furthermore, the Gα activator CTX significantly induced ROS generation in guard cells of the WT Col-0 and *gcr1* mutants in the absence or presence of UV-B or ACC (Figure 3). In addition, the overexpression of *cGPA1* not only slightly but significantly induced ROS generation in guard cells of the WT Col-0 and *gcr1* mutants in the absence of UV-B or ACC, but also rescued the defect of the *gcr1-1* and *gcr1-2* mutants in UV-B- and ACC-induced ROS production in guard cells (Figure 3). These results were consistent with the stomatal responses of the WT Col-0, *gcr1* mutants, and *cGPA1* transgenic lines to CTX, UV-B, and ACC (Figure 2), further suggesting that GCR1 mediates both UV-B- and ethylene-induced stomatal closure by activating Gα subunit GPA1-dependent ROS production in guard cells. To confirm this notion, we next investigated the stomatal response to exogenous hydrogen peroxide (H_2_O_2_). As expected, exogenous H_2_O_2_ not only induced stomatal closure in the leaves of the WT Col-0, *gcr1* mutants, and *cGPA1* transgenic lines in the absence of UV-B or ACC but also rescued the defects of the *gcr1-1* and *gcr1-2* mutants in UV-B- and ACC-induced stomatal closure (Figure 4). Together, our results firmly show that GCR1 mediates both UV-B and ethylene guard cell signaling by activating GPA1-dependent ROS production in guard cells.

### 2.4. GCR1 Mediates UV-B- and Ethylene-Induced Stomatal Closure by Activating GPA1-Dependent NO Production in Guard Cells

Because NO also mediates both UV-B- and ethylene-induced stomatal closure by working downstream of GPA1 and H_2_O_2_ [40,48], we further examined the relationship among NO, GCR1, and GPA1 in UV-B and ethylene stomatal signaling. We first measured NO levels in guard cells of the WT Col-0, *gcr1* mutants, and *cGPA1* transgenic lines using the NO-specific fluorescent dye 4,5-diaminofluorescein diacetate (DAF-2DA) in the absence or presence of CTX, UV-B, and ACC. Similar to the effect of UV-B, ACC, and CTX on ROS production in guard cells of the WT Col-0, *gcr1* mutants, and *cGPA1* transgenic lines (Figure 3), both UV-B- and ACC-triggered NO generation in guard cells of the WT Col-0 was completely impaired in the *gcr1* mutants; the activation of Gα by CTX could significantly induce NO production in guard cells of the WT Col-0 and *gcr1* mutants in the either absence or presence of UV-B or ACC; the overexpression of *cGPA1* slightly induced NO production in guard cells of the WT Col-0 and *gcr1* mutants in the absence of UV-B or ACC and also rescued the defects of the *gcr1* mutants in the UV-B- and ACC-induced NO generation in guard cells (Figure 5). These results were consistent with the stomatal responses of the WT Col-0, *gcr1* mutants, and *cGPA1* transgenic lines to CTX, UV-B, and ACC (Figure 2), indicating that GCR1 mediates both UV-B- and ethylene-induced stomatal closure through activating GPA1-dependent NO generation in guard cells. This notion was also supported by the finding that the exogenous application of NO donor sodium nitroprusside (SNP) not only induced stomatal closure in the WT Col-0, *gcr1* mutants, and *cGPA1* transgenic lines in the absence of UV-B or ACC but also rescued the defects of the *gcr1* mutants in the UV-B- and ACC-induced stomatal closure (Figure 6).

### 2.5. Ethylene Functions Upstream of GCR1 to Transduce UV-B Signaling in Guard Cells

GCR1 mediates both UV-B- and ethylene-induced stomatal closure by working upstream of GPA1, ROS, and NO (Figure 1, Figure 2, Figure 3, Figure 4, Figure 5 and Figure 6), while ethylene and GPA1 mediate UV-B-induced stomata closing as well as the production of ROS and NO in guard cells [40,49]. These findings suggest that ethylene mediates UV-B guard cell signaling via working upstream of GCR1. To test this suggestion, we examined the stomatal response of *gcr1* mutants to ACC in the presence of UV-B, as well as the effect of UV-B on ethylene production in *gcr1* mutants. As expected, the exogenous application of ACC could not reverse the defect of the *gcr1* mutants in UV-B-induced stomatal closure (Figure 7A); UV-B-induced ethylene production in the WT Col-0 leaves was also not affected by the *gcr1-1* and *gcr1-2* mutants (Figure 7B). These results support that ethylene functions upstream of GCR1 to transduce UV-B signaling in guard cells.

## 3. Discussion

In animals, extracellular signals are often perceived by GPCRs and transduced through G proteins to downstream targets. In plants, G protein signaling plays essential roles in diverse biological processes, including guard cell signaling, but little is known about their upstream receptors [50]. In Arabidopsis, only GCR1 was identified as a canonical GPCR gene [2,3] and bioinformatically analyzed to be the best GPCR candidate [7]. Furthermore, GCR1 is a potential prototypical GPCR in plants, because it directly binds to the plant Gα subunit GPA1 [8] like that for animal GPCRs [16]. However, the unsettled controversy over the existence of or the need for the prototypical, animal model GPCRs in plants has overshadowed a more fundamental quest for the role of GCR1 in plant G protein signaling [32,33]. In this study, by genetic and pharmacological analyses, we provided convincing evidence that GCR1 positively mediates UV-B- and ethylene-induced stomatal closure by activating GPA1-dependent ROS and NO production in guard cells. Evidence is also provided to show that ethylene acts upstream of GCR1 to transduce UV-B signaling in guard cells. This study establishes the existence of a classic paradigm of G protein signaling in guard cell signaling to UV-B and ethylene.

In guard cell signaling, Gα subunit GPA1 mediates many factor-triggered stomatal movements by controlling the production of ROS and NO and regulating the activities of ion channels in guard cells [34,35,36,37,38,39,40,41,42,43,44]. However, by what mechanisms many stimuli activate G protein signaling and whether GCR1 participates in the activation of G protein signaling in guard cells is still unknown. In this report, we found that the Arabidopsis T-DNA insertional null mutants for *GCR1* were insensitive to UV-B- and ethylene-induced stomatal closure as well as ROS and NO production in guard cells (Figure 1, Figure 3 and Figure 5), while exogenous H_2_O_2_ and NO induced stomatal closure in the *gcr1* mutants in either the presence or absence of UV-B or ethylene (Figure 4 and Figure 6). This pharmacological and genetic evidence convincingly indicates that GCR1 mediates both UV-B- and ethylene-induced stomatal closure by regulating ROS and NO generation in the guard cells of Arabidopsis. This role of GCR1 is similar to the role of GPA1 in guard cell signaling triggered by UV-B and ethylene [40,41], suggesting that GCR1 and GPA1 function in the same pathway. This suggestion is consistent with the previous biochemical evidence that GCR1 physically interacts with GPA1 [8]. Moreover, the defect of *gcr1* mutants in the UV-B- and ethylene-induced ROS and NO production in guard cells and subsequent stomatal closure can be rescued by the application of Gα activator CTX or overexpression of a constitutively active form of Gα subunit GPA1 (cGPA1) (Figure 2, Figure 3, Figure 4, Figure 5 and Figure 6), further proving that GCR1 mediates both UV-B- and ethylene-induced ROS and NO production in guard cells and subsequent stomatal closure by the activation of Gα subunit GPA1. This finding not only indicates the role of GCR1 in guard cell signaling to UV-B and ethylene but also proves the existence of a classic paradigm of G protein signaling in plants. However, the experimental identification of a ligand for GCR1 and demonstration of GEF activity of GCR1 would unequivocally establish this classic paradigm of G protein signaling in UV-B or ethylene signaling. Further work is required in this direction.

Based on analysis of the stomatal response of *GCR1* mutants *gcr1-3* and *gcr1-4*, Pandey and Assmann [8] indicated that GCR1 has a negative role in ABA and S1P guard cell signaling, as these *gcr1* mutants exhibit hypersensitivity in ABA and S1P inhibition of stomatal opening and promoting of stomatal closure. Here, our data of two other allelic *GCR1* mutants, *gcr1-1* and *gcr1-2* (Figure 1 and Appendix A), not only confirm the results of Pandey and Assmann [8] but further indicate the positive role of GCR1 in UV-B and ethylene guard cell signaling. Because the ABA and S1P responses of *gcr1* mutant guard cells are opposite to those of *gpa1* mutants [8,34,44], it is hypothesized that GCR1 negatively regulates ABA and S1P signaling via a mechanism dependent on its binding to and negatively regulating GPA1 or independent of its binding to GPA1 [8]. However, the directly negative regulation of GPA1 by GCR1 is opposite to the classic paradigm of G protein signaling and lacks supporting biochemical and genetic evidence; thus, it is possible that GCR1 negatively regulates ABA and S1P guard cell signaling via a mechanism independent of GPA1. Combined with our results, it is clear that the role and relationship of GCR1 and GPA1 in UV-B and ethylene guard cell signaling are much different from those in ABA and S1P guard cell signaling. Previous studies have shown that in the absence of ABA, ethylene can induce stomatal closure [51], but in the presence of ABA, it inhibits ABA-induced stomatal closure [52]. Thus, whether these complex effects of ethylene on stomatal movement are caused by the different roles of GCR1 in ABA and ethylene guard cell signaling is an interesting question to be studied in the future.

Our previous studies have shown that both the UV-B-specific signaling pathway UV RESISTANCE LOCUS 8 (UVR8)-CONSTITUTIVELY PHOTOMORPHOGENIC1 (COP1)-ELONGATED HYPOCOTYL5 (HY5)-dependent ethylene and GPA1 mediate UV-B-induced stomatal closing via functioning upstream of ROS and NO [40,49] and that GPA1 mediates ethylene-triggered stomatal closure via working downstream of the ethylene receptor ETHYLENE RESPONSE 1 (ETR1) and the ethylene signaling negative regulator CONSTITUTIVE TRIPLE RESPONSE 1 (CTR1) and upstream of ROS and NO [41,42]. Here, we further show that GCR1 mediates both UV-B- and ethylene-induced stomatal closure by working upstream of GPA1, ROS, and NO (Figure 1, Figure 2, Figure 3, Figure 4, Figure 5 and Figure 6). These results suggest that ethylene acts as a secondary messenger to mediate UV-B guard cell signaling via working downstream of the UVR8-COP1-HY5 signaling pathway and upstream of GCR1, which is further supported by the presented evidence that *gcr1* mutants show insensitivity to ethylene induction of stomatal closure in the presence or absence of UV-B but sensitivity to UV-B induction of ethylene production (Figure 7).

In summary, the data presented here, combined with those of previous studies, provide compelling evidence for the essential role of GCR1, a potential prototypical GPCR in Arabidopsis, in UV-B and ethylene guard cell signaling and establish potential models of the UV-B- and ethylene-signaling pathways in guard cells (Figure 8). UV-B signaling in guard cells is initiated by the UV-B-specific signaling pathway UVR8-COP1-HY5 via the induction of ethylene production [48]. Ethylene is perceived by its receptor, ETR1, to inactivate the CTR1 kinase [40], releasing GCR1-dependent GPA1 activation (Figure 2). Then activated, GPA1 induces ROS and NO generation in guard cells and subsequent stomata closing (Figure 3, Figure 4, Figure 5, Figure 6 and Figure 8). However, the bacterial toxin CTX, which activates the Gα subunit as those of non-hydrolyzing GTP analogues that induce a conformational change in the Gα subunit that promotes the dissociation of the trimeric complex [37], could significantly induce ROS and NO production in guard cells and subsequent stomatal closure in WT and *gcr1* mutants in either the absence or presence of UV-B or ACC (Figure 2, Figure 3 and Figure 5), while the overexpression of cGPA1, a potential constitutively active form of GPA1 that was shown to disable the GTPase activity of Gα once activated [53], only slightly induced ROS and NO generation and subsequent stomatal closure in the WT and *gcr1* mutants in the absence of UV-B or ACC, but completely rescued the defects of the *gcr1* mutants in UV-B- and ACC-induced ROS and NO production in guard cells and stomatal closure (Figure 2, Figure 3 and Figure 5). These different effects between CTX and cGPA1 suggest that the complete activation of this overexpressed cGPA1 still requires the presence of UV-B or ACC. It is possible that the active site of the cGPA1 is suppressed by the formation of heterotrimeric G protein complexes [47], and that the dissociation of cGPA1 from the trimeric complex, and thus its complete activation, requires other UV-B- or ethylene-triggered signals. Many studies have shown that plant G proteins can physically interact with atypical receptors such as the Arabidopsis RGS1 and some RLKs, which promote the dissociation of the heterotrimeric G protein complex, and thus Gα activation, by phosphorylation or by signals inhibiting the deactivation of a constitutively active Gα [15,22,23,24,25,26,27,28,29,30,31,32,33]. Combined together, it is proposed that both the classic GPCR-dependent and the atypical GPCR-independent G protein signaling may work cooperatively to control the stomatal response to UV-B and ethylene (Figure 8), but this suggestion should be studied in the future.

## 4. Materials and Methods

### 4.1. Plant Materials and Growth Conditions

*Arabidopsis thaliana* seeds were obtained from the Nottingham Arabidopsis Stock Center (Nottingham, UK). Ecotype Col-0 was used as the WT control, mutants of *gcr1-1* (N6539) and *gcr1-2* (N6540) are in the Col-0 ecotype background, and their genotypes were confirmed by PCR. The seeds were sterilized, sown in a potting mix of soil: vermiculite: perlite (3:1:1, *v/v/v*), and grown in plant growth chambers under a 16 h photoperiod (0.1 mmol/m^2^/s) at a day/night temperature cycle of 22/18 °C and relative humidity of 80%. Fully expanded rosette leaves of 4-week-old plants were harvested and used for analysis.

### 4.2. UV-B Treatments

In this paper, 0.5 W/m^2^ UV-B radiation was used for all UV-B treatments and carried out, as previously described [40,54].

### 4.3. Stomatal Bioassays

Freshly detached leaves with their abaxial surfaces facing up were first incubated in MES-KCl buffer (10 mmol/L MES, 50 mmol/L KCl, and pH 6.15) under white light (0.1 mmol/m^2^/s) for 2 h to open stomata and then transferred to the MES-KCl buffer with or without 100 μmol/L ACC, 10 μmol/L ABA, 400 ng/mL CTX, 100 μmol/L H_2_O_2_, or 100 μmol/L SNP under white light alone or supplemented with 0.5 W/m^2^ UV-B radiation for another 3 h. The pH of the MES-KCl buffer containing ACC was adjusted to 6.15 with KOH to avoid any effect of the pH. After the treatments, the abaxial epidermis was immediately peeled from the treated leaves, and the stomatal apertures were recorded using a calibrated light microscope.

### 4.4. Measurement of Leaf Water Loss Rate

For the measurement of leaf water loss, ten freshly detached leaves with open stomata were incubated in MES-KCl buffer alone or containing 100 μmol/L ACC under white light (0.1 mmol/m^2^/s) alone or supplemented with 0.5 W/m^2^ UV-B radiation for 3 h, weighed and labeled as W_0_. The treated leaves were then placed under the same light conditions without MES-KCl buffer for 2 h, weighed and labeled as W_1_. The leaf water loss rate was calculated as W_0_–W_1/_W_0_ × 100.

### 4.5. Measurement of ROS and NO Levels in Guard Cells

The fluorescent probes H_2_DCFDA and DAF-2DA were used to detect levels of ROS and NO in guard cells, respectively, as previously mentioned [48,54]. Briefly, the freshly prepared epidermal strips peeled from the abaxial surfaces of leaves were incubated in Tris-KCl loading buffer (10 mmol/L Tris and 50 mmol/L KCl; pH 7.2) in the presence of 50 μmol/L H_2_DCF-DA for 10 min or 10 μmol/L DAF-2 DA for 30 min under darkness. After washing in fresh Tris-KCl buffer under darkness, an inverted fluorescence microscope (Eclipse TE 200; Nikon) with an excitation filter (450–490 nm) and an emission filter (520–560 nm) was used to detect H_2_DCF and DAF-2 fluorescence in guard cells. In each experiment, five epidermal strips taken from the leaves of different plants were examined, and each experiment was repeated at least three times. The fluorescence pixel intensity was analyzed in the whole stomata areas of the fluorescence microscope images with Leica Image software (Leica Microsystems), and the data are presented as means ± SE of three independent experiments, each with 20 stomata. The presented images are representative of the three or more independent experiments performed.

### 4.6. Generation of Transgenic Plants Overexpressing cGPA1

The generation of transgenic lines overexpressing cGPA1 was based on the protocol described [47], with some modifications. A cDNA library was generated from RNA isolated from seedlings of Arabidopsis ecotype Col-0. The wild-type GPA1 cDNA was amplified by PCR using this cDNA library as a template with gene-specific primers GPA1 KpnI+ (5′-GGTACCATGGGCTTACTCTGCAGTAGAA-3′) and GPA1 SmaI- (5′-TCCCCCGGGTAAAAGGCCAGCCTCCAGTAA-3′). The cGPA1 was generated by changing Glu-222 to Leu by PCR–based mutagenesis using primers M1 (5-TGACGTGGGTGGACTGAGAAATGAGAGGAGG-3) and M2 (5-CCTCCTCTCATTTCTCAGTCCACCCACGTCA-3) in combination with the primers GPA1 KpnI+ and GPA1 SmaI-. These fragments were inserted into *Sma*I and *Kpn*I sites in the pCambia1305 vector, resulting in 35S:*cGPA1*, and transferred into *A. tumefaciens* strain GV3101 cells, which were then used to transform the Col-0, *gcr1-1*, or *gcr1-2* plants by using the floral dip method. Seedlings were selected on 1/2 Murashige and Skoog solid medium containing 40 μg/mL hygromycin-B, and T3 transgenic plants were used for the experiments.

### 4.7. RNA Extraction and qPCR

Total RNA was extracted from the treated leaves using Magzol^TM^ reagent (Trizol reagent, Magen). An amount of 1 μg of total RNA was used to synthesize cDNA using the Prime Script™ RT Master Mix (Takara Bio), according to the manufacturer’s instructions. qPCR analysis was performed with SYBR Green TaqMix (Takara Bio) on the IQ5 real-time system (Bio-Rad). Each qPCR result was the average of three independent biological repeats. The relative expression level is shown as a value relative to the WT Col-0 sample after normalization to that of *ACTIN 2*. The sequences of the primers for qPCR are *GCR1*-*F* (5′-TCTGCGTTGCTTCGTTCTTGT-3′), GCR1-R (5′-TTTCGCTTCGCTGTTGTTGTT-3′), *ACTIN2-F* (5′-CAAGGCCGAGTATGATGAGG-3′), and *ACTIN2-R* (5′-GAAACGCAGACGTAAGTAAAAAC-3′).

### 4.8. Ethylene Measurement

After the treatments, the leaves were enclosed in a vial. After 3 h, the ethylene levels in the vial were analyzed, as previously mentioned [49]. For each treatment, five leaves from different plants were measured, and the treatment was repeated at least four times. Data are shown as means ± SE of three independent experiments.

### 4.9. Statistical Analysis

The samples were arranged in completely randomized designs with three replications. The data were presented as the mean value ± standard errors (SE). Results from different treatments were compared using one-way ANOVA (analysis of variance). Following ANOVA, *post hoc* comparisons of means were made using Mann–Whitney multiple comparisons. Statistical significance was determined at *p* < 0.01 or *p* < 0.05, as indicated in the figure legends. Statistical analyses were carried out using SPSS16.0.

## Figures and Tables

**Figure 1 ijms-23-05512-f001:**
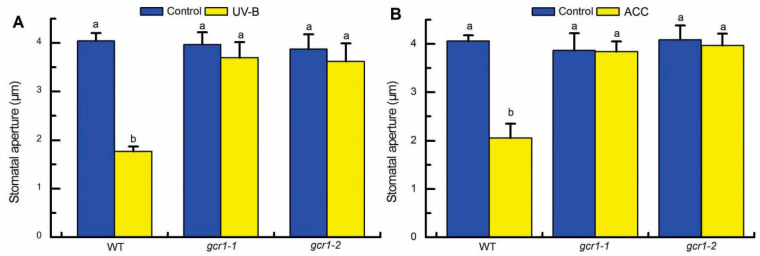
GCR1 plays a positive role in UV-B and ethylene induction of stomatal closure. (**A**) Leaves of the wild-type Col-0 (WT) and *gcr1* mutants with open stomata were exposed to light alone (Control) or with 0.5 W/m^2^ UV-B (UV-B) for 3 h. (**B**) Leaves of the WT and *gcr1* mutants with open stomata were incubated in MES buffer alone (Control) or containing 100 μmol/L ACC (ACC) for 3 h under the light. After treatments, stomatal apertures were measured in epidermal strips taken from the abaxial surfaces of the treated leaves. Data are presented as means ± SE of three independent experiments, each with 30 stomata. Means with different letters are significantly different at *p* < 0.01.

**Figure 2 ijms-23-05512-f002:**
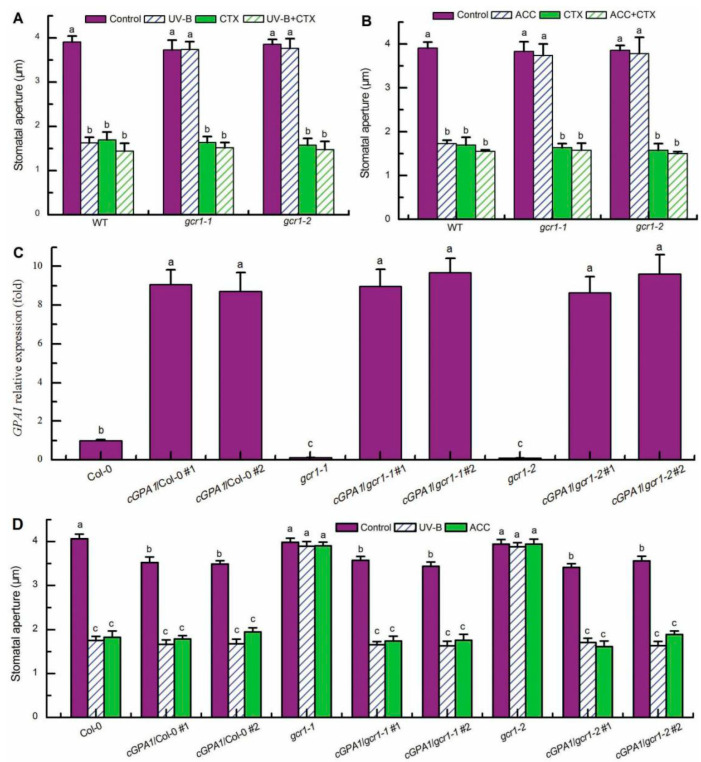
GCR1 mediates UV-B- and ethylene-induced stomatal closure via activating Gα subunit GPA1. (**A**) Leaves of the wild-type Col-0 (WT) and *gcr1* mutants with open stomata were floated on MES buffer under light alone (Control) or with 0.5 W/m^2^ UV-B (UV-B) or the MES buffer containing 400 ng/mL CTX (CTX) for 3 h. (**B**) Leaves of the WT and *gcr1* mutants with open stomata were incubated in MES buffer in the absence (Control) or presence of 100 μmol/L ACC (ACC) or 400 ng/mL CTX (CTX) for 3 h under the light. (**C**) qPCR to assess *GPA1* relative expression level in the leaves of T2 transgenic lines overexpressing *c**GPA1* in the background of wild-type Col-0 and the *gcr1-1* and *gcr1-2* mutants. The transcript level of *GPA1* in Col-0 was set to one. Data are shown as means ± SE of three independent biological determinations, and means with different letters are significantly different at *p* < 0.01. (**D**) Leaves of the WT Col-0, *gcr1* mutants, and transgenic lines *cGPA1*/Col-0/*gcr1-1*/*gcr1-2* with open stomata were floated on MES buffer under light alone (Control) or with 0.5 W/m^2^ UV-B (UV-B) or the MES buffer containing 100 μmol/L ACC (ACC) for 3 h. For (**A**), (**B**), and (**D**), stomatal apertures were measured in epidermal strips taken from the abaxial surfaces of the treated leaves. Data are shown as means ± SE (*n* = 3), each with 30 stomata. Means with different letters are significantly different at *p* < 0.01.

**Figure 3 ijms-23-05512-f003:**
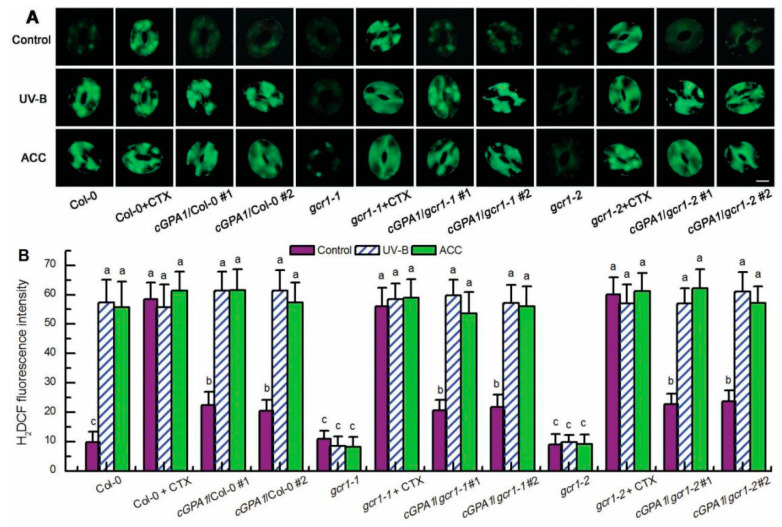
GCR1 mediates UV-B- and ethylene-induced ROS production in guard cells via activating Gα subunit GPA1. Leaves of the wild-type Col-0, *gcr1* mutants, and transgenic plants *cGPA1*/Col-0/*gcr1-1*/*gcr1-2* were floated on MES buffer in the absence or presence of 400 ng/mL CTX or 100 μmol/L ACC (ACC) and exposed to light alone (Control) or with 0.5 W/m^2^ UV-B (UV-B) for 3 h. Then the images (**A**) and fluorescence intensities (**B**) of guard cells preloaded with H_2_DCFDA were recorded. For (**A**), scale bar = 10 μm. For (**B**), data are shown as means ± SE (*n* = 3), each with 20 stomata. Means with different letters are significantly different at *p* < 0.05.

**Figure 4 ijms-23-05512-f004:**
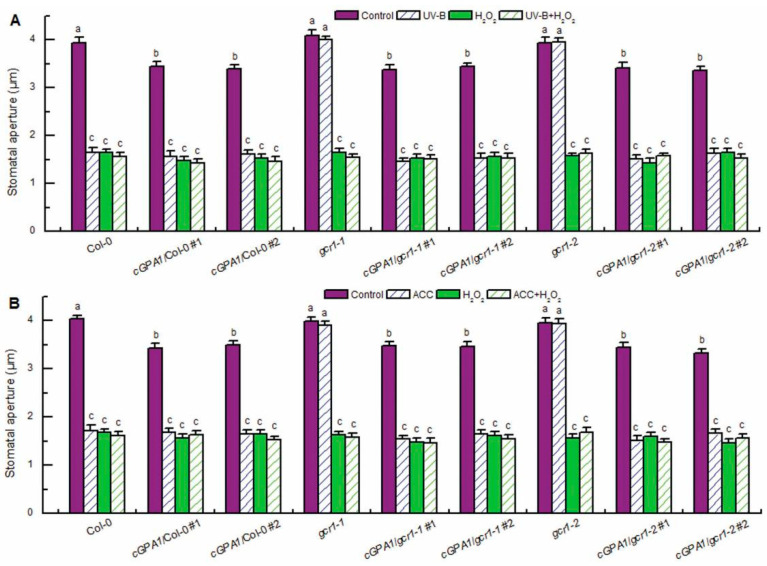
Exogenous H_2_O_2_ induced stomatal closure in the *gcr1* mutants and *c**GPA1* transgenic plants in the absence or presence of UV-B or ACC. Leaves of the wild-type Col-0, *gcr1* mutants, and transgenic plants *cGPA1*/Col-0/*gcr1-1*/*gcr1-2* with open stomata were floated on MES buffer in the absence or presence of 100 μmol/L H_2_O_2_ (H_2_O_2_) under light alone (Control) or with 0.5 W/m^2^ UV-B (UV-B) (**A**) or the MES buffer containing 100 μmol/L ACC (ACC) (**B**) for 3 h. Then stomatal apertures were measured in epidermal strips. Data are shown as means ± SE (*n* = 3), each with 30 stomata. Means with different letters are significantly different at *p* < 0.01.

**Figure 5 ijms-23-05512-f005:**
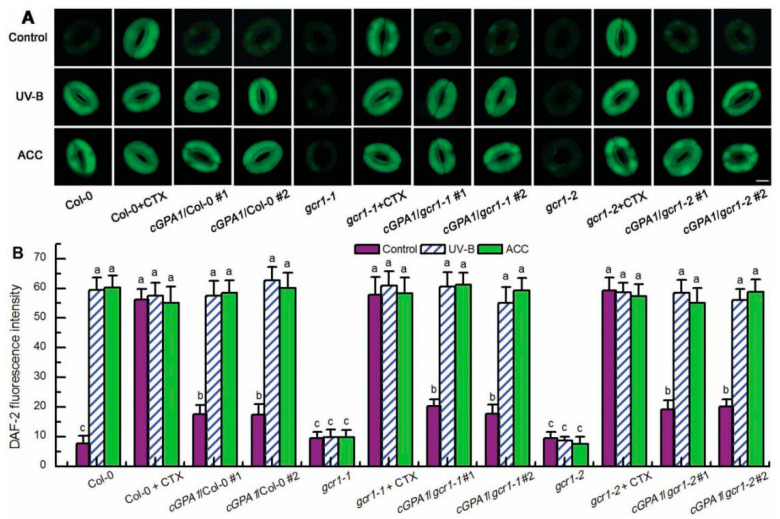
GCR1 mediated UV-B- and ethylene-induced NO generation in guard cells via activating Gα subunit GPA1. Leaves of the wild-type Col-0, *gcr1* mutants, and transgenic plants *cGPA1/Col-0/gcr1-1/gcr1-2* were floated on MES buffer in the absence or presence of 400 ng/mL CTX (CTX) or 100 μmol/L ACC (ACC) and exposed to light alone (Control) or with 0.5 W/m^2^ UV-B (UV-B) for 3 h. Then the images (**A**) and fluorescence intensities (**B**) of guard cells preloaded with DAF-2DA were recorded. For (**A**), scale bar = 10 μm. For (**B**), data are shown as means ± SE (*n* = 3), each with 20 stomata. Means with different letters are significantly different at *p* < 0.05.

**Figure 6 ijms-23-05512-f006:**
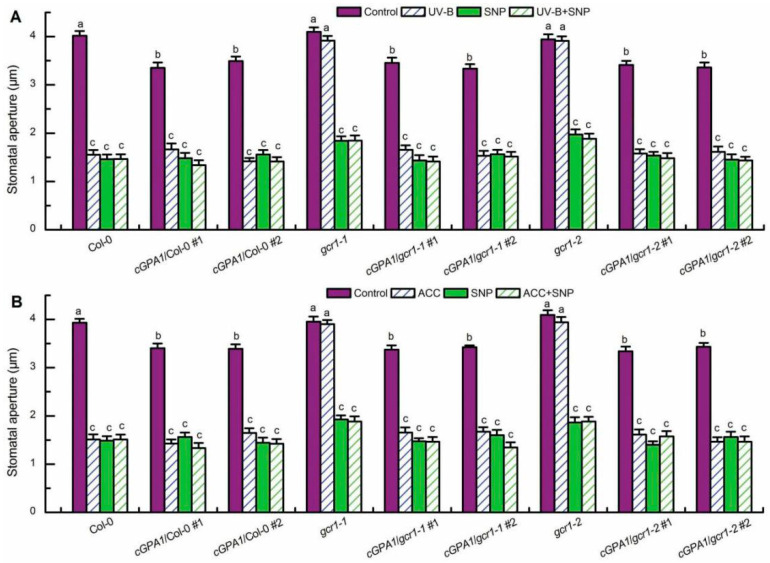
Exogenous NO induced stomatal closure in the *gcr1* mutants and *c**GPA1* transgenic plants in the absence or presence of UV-B or ACC. Leaves of the wild-type Col-0, *gcr1* mutants, and transgenic plants *cGPA1*/Col-0/*gcr1-1*/*gcr1-2* with open stomata were floated on MES buffer in the absence or presence of 100 μmol/L SNP (SNP) and exposed to light alone (Control) or with 0.5 W/m^2^ UV-B (UV-B) (**A**) or the MES buffer containing with 100 μmol/L ACC (ACC) (**B**) for 3 h. Then stomatal apertures were measured in epidermal strips. Data are shown as means ± SE (*n* = 3), each with 30 stomata. Means with different letters are significantly different at *p* < 0.01.

**Figure 7 ijms-23-05512-f007:**
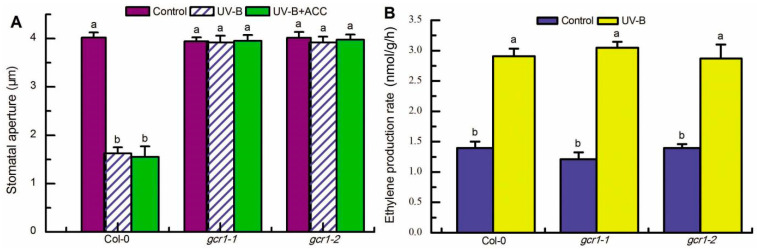
Ethylene acted upstream of GCR1 in UV-B guard cell signaling. Leaves of the wild-type Col-0 and *gcr1* mutants were floated on MES buffer in the absence or presence of 100 μmol/L ACC (ACC) and exposed to light alone (Control) or with 0.5 W/m^2^ UV-B (UV-B) for 3 h. Then the stomatal apertures were measured in the epidermal strips (**A**), or ethylene production rate was analyzed in the treated leaves by gas chromatography (**B**). Data are shown as means ± SE (*n* = 3), and means with different letters are significantly different at *p* < 0.01.

**Figure 8 ijms-23-05512-f008:**
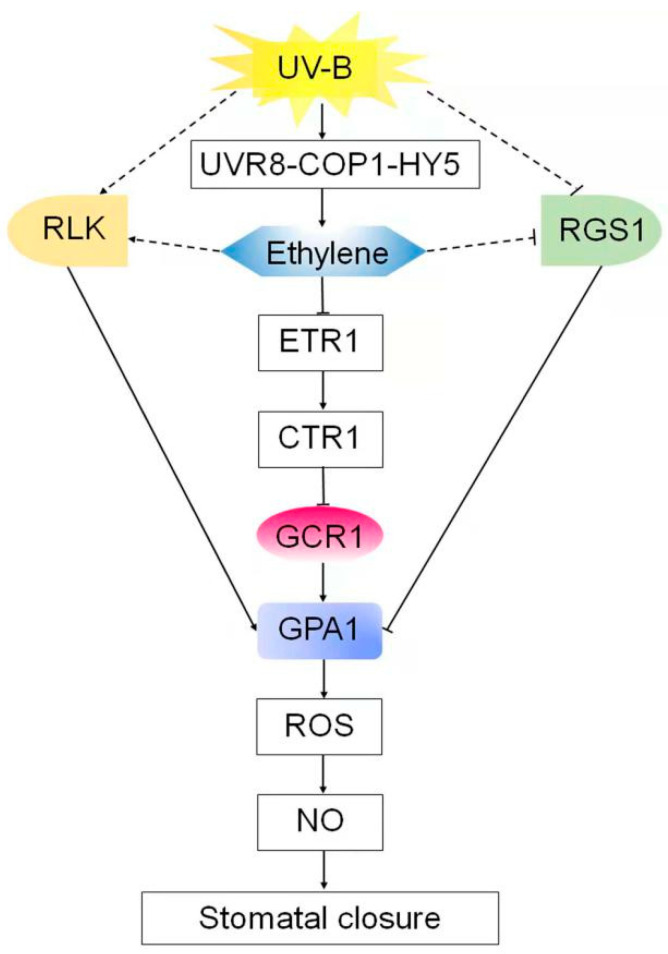
Modules showing the possible signaling pathways for UV-B- and ethylene-induced stomata closing. UV-B signaling in guard cells is initiated by the UV-B-specific signaling pathway UVR8-COP1-HY5 via induction of ethylene production. Ethylene is perceived by its receptor, ETR1, to inactivate the CTR1 kinase, releasing GCR1-dependent GPA1 activation. Then activated, GPA1 induces ROS and NO generation in guard cells and subsequent stomata closing. UV-B and ethylene maybe also activate GPA1 by inducing receptor-like kinases (RLKs)-phosphorylated GPA1 or by inhibiting RGS1-dependent deactivation of GPA1. Arrows and bars indicate induction and inhibition, respectively. The dashed line indicates a hypothetical cell response.

## Data Availability

Data will be made available upon request.

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
