# Peer review of "GCR1 Positively Regulates UV-B- and Ethylene-Induced Stomatal Closure via Activating GPA1-Dependent ROS and NO Production"

_ijms, 2022, doi:10.3390/ijms23105512_

Round 1

Reviewer 1 Report

In the manuscript authors showed that, gcr1 and gcr2 mutants failed in UV-B- and ethylene-induced stomatal closure and hydrogen peroxide (H2O2) and nitric oxide (NO) production in guard cells. Application of Gα activator and overexpression of a constitutively active form of Gα subunit GPA1 (cGPA1) are able to rescue the defective phenotype of stomatal aperture regulation. Via measuring stomatal aperture after different treatments of ACC, UV-B, CTX, H2O2, SNP, & combinations of these, and checking internal H2O2 and NO changes, authors showed that GCR1 positively controls UV-B- and ethylene-induced stomatal closure by activating GPA1-dependent H2O2 and NO production in guard cells, and also suggesting that ethylene acts upstream of GCR1 to transduce UV-B guard cell-signaling.

There are few comments regarding this manuscript mentioning below,

  1. Line 43, 53-54: The gene names explaining first time should have the expanded name of the genes. Please follow this while explaining a gene first time.
  2. While explaining about UV-B mediated stomatal regulation, there are UV-B related signaling components should be discussed (e.g. HY5, COP1, etc.), and it will be good to functionally validate the role of G-proteins with UV-B signaling elements, that can provide more information in this aspect.
  3. It is important to confirm the ethylene and UV-B induced GCR1-GPA1 interaction. The interaction GCR1-GPA already reported, whereas experiment showing ethylene and UV-B induced interaction will provide strong support to the results. For example, pull down assay after treatment. (Reproduce the interaction between GCR1-GPA1).
  4. The functional validation of gpa1 mutants with respect to UV-B and ethylene mediated stomatal closure, and the validation of gpa1gcr1 and gpa1gcr2 double mutants will provide functional relation in stomatal regulation.
  5. Provide possible reasons behind the differential responses of Gα activator treated and overexpression (cGPA1) plants in NO production, because both leading to similar function.

Reviewer 2 Report

The manuscript presented by Li et al. is focused on the role of putative GCR1 in stomatal closure induced by UV-B and ethylene. The authors show that the mutant gcr1 is impaired in UV-B and ethylene induced stomatal closure. The authors have used pharmacology and genetic approaches to show that the activated G protein GPA1 can rescue stomata closure in the gcr1 mutant. The signaling molecules H2O2 and NO were also investigated.

The manuscript is well presented and easy to follow. However, a summary of signaling cascade could be added as figure to help the readers understand the finding. The paper sometime lack more details. My concerns are:

  1. H2DCFDA is not a H2O2 sensor dye, but a cumulative ROS dye. Please correct “H2O2” in the corresponding sentences.
  2. Fluorescence quantification is not explained and it is difficult to conclude if the procedure is correct. Is the fluorescence level an average of stomata or leaves? Authors should indicate how they proceeded. A brief explanation is a minimum instead of citing other articles.
  3. There are no details on acquisition parameters and image analysis
  4. The authors overexpressed the constitutive active GPA1 in the plant with the 35S promoter. I wonder why they didn't choose a guard cell promoter? There is no comment on the potential phenotype due to overexpression of constitutive active GPA1. What do the leaves look like?
  5. 5) The gpa1 and 35S-GPA1-Q222L mutants exhibit a phenotype (density of leaves and stomata). The authors did not comment on the potential phenotype of their cGPA1 line in the gcr1 context (stomata density, size etc.). In addition, some measurements of leaf water loss or stomatal opening in situ must be given to confirm the authors' data on the epidermal strip.
  6. Bioassays on stomata are easy to perform. But I wonder about the approach of the authors. Peeling the strip of epidermis is stressful and I wonder how the authors did not observe any ROS fluorescence in the control condition and how the stomatal opening can be preserved.
  7. Stomatal bioassays are usually blinded or double-blinded. Are the authors proceeding blind?
  8. Could the author provide which statistic test they performed? Is it not clear which post-hoc test they did?

Reviewer 3 Report

The work entitled “GCR1 positively regulates UV-B- and ethylene-induced stomatal closure via activating GPA1-dependent H2O2 and NO production” is a very well written manuscript, presenting results of pharmacological and genetic studies on Arabidopsis thaliana GCR1 protein involvement in the regulation of UV-B and ethylene-induced stomatal closure via activating the Gα subunit of the heterotrimeric G protein (GPA1) – dependent H2O2 and NO production.

Dear All,

Please, find my comments and suggestions.

  1. Introduction:

Line 50

“Thus, the existence of these prototypical, animal model, GPCRs in plants is still an unsettled controversy [7,9].”

This sentence is not grammatically correct.

Line 52-55

I suggest to add short information about the three extra-large Gα subunits (XLG1, XLG2 and XLG3).

Line 64-67

“Thus, it has been proposed that, instead of the GPCR-dependent activation of G-proteins used in animals, plant G-proteins are activated by phosphorylation or by signals inhibiting the deactivation of a constitutively active Gα [31,32].”

I suggest to use other term than “used”

Line 67-69

I suggest to rewrite this sentence and possibly split it into two, e.g. after GPCR you could place a dot, and then you could start the second sentence with “However, whether…”

Line 92-93

The sentence is not complete.

Line 99

Instead of “via” I propose “in”

Results

Results 2.1.

Line 121

I suggest to remove “with the two other allelic GCR1 mutants gcr1-1 and gcr1-2

Line 122

I suggest to write “Consistent with the results of Pandey and Assmann [8] that gcr1-3 and gcr1-4 (two other allelic GCR1 mutants) exhibited”

Line 124

I suggest to write “our results showed that gcr1-1 and gcr1-2 are hypersensitive in both ABA responses”

Line 128

Please, remove the unused space.

Results 2.2.

Line 149

Please, write the names of mutants (here, and in other places) in italics.

Line 157

I believe you wanted to write just (cGPA1) not (cGPA1):GPA1.

Line 159

“We then analyzed”, not “;we then analyzed”

Line 174

Please, write about the samples that were used for GPA1 relative expression level assessment (type of tissue).

Please, try to refer to the differences in the GPA1 expression levels in WT vs gcr1-1 and gcr1-2.

Results 2.3.

Line 189

“Then the images” not “, then the images”. The same in the text describing other figures.

Line 205

There I some problem with lines.

Discussion

 Line 291

I suggest to write “In Arabidopsis, only GCR1 was identified …”

Line 293

Please, rewrite this sentence as it is not grammatically correct.

I suggest to refer to the BR-induced expression of AtACS5 and AtACS9 to initiate ethylene synthesis, which signals through Gα to synthesize H2O2.

(Shi, C.; Qi, C.; Ren, H.; Huang, A.; Hei, S.; She, X. Ethylene Mediates Brassinosteroid-Induced Stomatal Closure via Gα Protein-Activated Hydrogen Peroxide and Nitric Oxide Production in Arabidopsis. Plant J. 2015, 82, 280–301)

Materials and methods

Materials and methods 4.1

 I suggest to add a supplementary figure that depicts the locus of interest along with locations of T-DNA insertions in gcr1-1 and gcr1-2 (possibly for gcr1-3 and gcr1-4 a well) mutants.

Have you checked the number of T-DNA insertions in gcr1-1 and gcr1-2 mutants? Are you sure that these lines carry T-DNA insertion only in the gene of interest?

Materials and methods 4.3

Are you sure that in the case of other buffers the pH was not changed?

Materials and methods 4.5

Can you elaborate more on the way the site-specific mutation was created?

Materials and methods 4.6

I believe the RNA was not isolated from treated samples.

Materials and methods 4.7

Can you write how many samples were used for ET measurements?

Round 2

Reviewer 1 Report

Authors have responded most of points. It is now suitable for publication.